# Vegetarian and Vegan Diets: Beliefs and Attitudes of General Practitioners and Pediatricians in France

**DOI:** 10.3390/nu14153101

**Published:** 2022-07-28

**Authors:** Cécile Villette, Pauline Vasseur, Nathanael Lapidus, Marion Debin, Thomas Hanslik, Thierry Blanchon, Olivier Steichen, Louise Rossignol

**Affiliations:** 1Département de Médecine Générale, Université de Paris Cité, 16 rue Henri Huchard, F75018 Paris, France; villette.cecile@gmail.com; 2Institut Pierre Louis d’Epidémiologie et de Santé Publique, Sorbonne Université, INSERM, IPLESP, 27 rue de Chaligny, F75012 Paris, France; pauline.vasseur2@gmail.com (P.V.); nathanael.lapidus@inserm.fr (N.L.); marion.debin@iplesp.upmc.fr (M.D.); thomas.hanslik@aphp.fr (T.H.); thierry.blanchon@upmc.fr (T.B.); 3Public Health Department, Saint-Antoine Hospital, AP-HP, Sorbonne Université, F75012 Paris, France; 4UFR Simone Veil-Santé, Université Versailles Saint Quentin en Yvelines, 55 Avenue de Paris, F78000 Versailles, France; 5AP-HP, Hôpital Ambroise Paré, Service de Médecine Interne, 9 Avenue Charles de Gaulle, F92100 Boulogne Billancourt, France; 6AP-HP, Hôpital Tenon, Service de Médecine Interne, Sorbonne Université, 4 rue de la Chine, F75020 Paris, France; olivier.steichen@aphp.fr

**Keywords:** vegetarian diet, family medicine, attitudes and practices in health

## Abstract

Studies suggest a decreasing trend in the consumption of meat products and a growing interest in vegetarian diets. Medical support may be relevant, especially when switching to a vegan diet. Our objective was to describe the beliefs and attitudes of primary care physicians toward vegetarian diets. A cross-sectional survey was conducted among general practitioners and pediatricians thorough a questionnaire including socio-demographic characteristics, specific care to vegetarians, and the risks and benefits of vegetarian diets according to physicians. Out of the 177 participating physicians, 104 (59%) have seen at least one vegetarian patient in consultation in the last three months. Half of the physicians declared that they would dissuade their patients from switching to a vegan diet (*n* = 88, 51%) and 14% (*n* = 24) from switching to an ovo-lacto-vegetarian (OLV) diet. Most physicians (*n* = 141, 88%) did not feel informed enough about these diets. Physicians thought that the most frequent deficiencies for OLV and vegan diets were iron (76% and 84%, respectively) and protein (45% and 79%, respectively). These results highlight the fact that French primary care physicians feel concerned by this subject and need more information on these diets. Specific recommendations would be useful to support their practice and relationship with vegetarians.

## 1. Introduction

In recent studies, vegetarians, including ovo-lacto-vegetarian (OLV) and vegan diets, represented 2% of the population in the United States, 3.25% in the United Kingdom, and 32.8% in India [1,2,3]. In France, the proportion of vegetarians has been estimated to represent 2.7% of the population [4]. Several French studies suggested a downward trend in the consumption of meat products: the average yearly consumption of meat decreased from 94 kg in 1998 to 86 kg in 2014 [5]; sales of vegetarian and vegan products increased by 24% in 2018 [6].

Evidence about the potential benefits of vegetarian diets (including the vegan diet) is inconsistent and confounded by many factors. A protective effect of the vegetarian diets was found on blood pressure as well as on type 2 diabetes and obesity, even in childhood [7,8,9,10,11,12]. Regarding cancer, the results of studies are inconsistent, except for the consumption of red meat (especially of processed meat), which is associated with an increased risk of cancer (especially colorectal and stomach) [13,14,15]. For stroke, myocardial infarction, and overall mortality, the results are inconsistent [16,17,18].

Regarding the risks associated with these diets, vitamin B12 deficiency is the only one that has been clearly described, especially for the vegan diet [19]. This deficiency slowly occurs and may remain undetected until its clinical consequences show up. A recent systematic review showed that a vegan diet may result in deficiencies in micronutrients (vitamin B12, zinc, calcium, and selenium), which should not be disregarded [20]. Another systematic review qualitatively comparing the vegetarian and nonvegetarian diets concluded that OVLs and vegans had a higher overall diet quality (4.5–16.4 points higher on the Healthy Eating Index 2010 (HEI-2010)) compared with nonvegetarians in 9 of 12 studies [21]. The authors explained that, for vegetarians, higher scores were driven by a closer adherence to recommendations for total fruit, whole grains, seafood and plant protein, and sodium. For nonvegetarians, adherence to recommendations for refined grains and total protein foods was better.

Diet is central to improving health outcomes. However, nutrition counseling is very limited during medical visits worldwide [22,23]. Limited time and gaps in knowledge have been identified as barriers [22,24]. Concerning the knowledge of diets, a systematic review showed that nutrition is insufficiently incorporated into medical education, regardless of country, setting, or year of education [25]. Deficits in nutrition education affect students’ knowledge, skills, and confidence to implement nutrition care into patient care. A study evaluated how U.S. medical students and family medicine residents rated their nutrition training [26]. Most trainees felt that they were inadequately trained during medical school, although they expressed a high interest in whole-foods plant-based nutrition to address the burden of chronic diseases [26].

Primary care physicians (general practitioners and pediatricians) are the first-care providers for the population. Our objective was to describe the beliefs and attitudes of general practitioners (GPs) and pediatricians toward the OLV and vegan diets.

## 2. Materials and Methods

### 2.1. Design and Participants

In June and July 2018, we led a cross-sectional survey among GPs and pediatricians of the French Sentinel Network. The Sentinel Network (http://www.sentiweb.fr (accessed on 27 July 2022) is a computerized system comprising 1300 volunteer GPs (about 2% of the French GPs population) and 114 pediatricians, located throughout mainland France and participating in the ongoing surveillance of health indicators and epidemiological studies [27]. We only contacted physicians who had participated in at least one survey since 2016.

### 2.2. Survey

The survey was created using LimeSurvey^®^ (LimeSurvey GmbH est représentée par Carsten Schmitz et Jory Nagel, HAmbourg, Deutschland) and sent by e-mail in June 2018. Two reminders were carried out in two-week intervals. We defined the OLV diet as a diet that excludes the consumption of meat (including fish, seafood, and the flesh of any other animal) but may include eggs, dairy, and honey; and we defined the vegan diet as one which excludes all animal products. The survey contained 31 questions and was split into four parts:Physician’s characteristics: demography, practice, location, nutrition degree, and personal diet.Care of vegetarian patients: impact on the physician–patient relationship, advice provided on switching to an OLV or vegan diet, and specific monitoring and treatments.Representations of the risks and benefits of the OLV and vegan diets. Physicians had to estimate: (i) the impact of an OLV or vegan diet on several health issues from 0 (“deleterious effect”) to 10 (“protective effect”) for adults and children; (ii) the risk of deficiencies associated with these two diets; (iii) whether the diets were suitable for different types of populations, including some specific subgroups (e.g., pregnant women, the elderly, infants, children, teenagers, athletes, and patients with diabetes or chronic kidney disease).Understanding of the patients following a vegetarian diet: perceived motives for the patients to adopt a vegetarian diet (health, environmental, ethical reasons, beliefs, fashion, or eating disorders).

In the survey, nutrition degrees included university diplomas and interuniversity diplomas in the field of nutrition.

### 2.3. Data Analysis

A descriptive analysis was performed on all of the studied variables. Outliers were checked and corrected if necessary. GP age was split into two groups (less than 50 years vs. 50 years old or more).

The determinants of physician’s advice against switching to a vegan diet were analyzed only for the GPs (the number of pediatricians was too small). For this analysis, the variables grading the strength of opposition and the perceived risks of deficiency were recoded into a binary variable (categories “strongly agree” and “agree” were grouped into “agree” and categories “disagree”, “strongly disagree”, and “I do not know” into “disagree”). The association between advising against switching to a vegan diet and each explanatory variable was tested in unadjusted logistic models. These variables were chosen based on the literature and clinical considerations. The odds ratios, with their 95% confidence interval, are given for each variable.

## 3. Results

### 3.1. Characteristics of the Physicians

Out of the 636 physicians of the Sentinel Network who were contacted for the survey (521 GPs and 115 pediatricians), 28% completed the questionnaire: they comprised 149 GPs and 28 pediatricians. The median ages of the responding GPs and pediatricians were 49 and 56 years old, respectively. Women accounted for 93% of the pediatricians (*n* = 26) and 41% of the GPs (*n* = 61) (Table 1). Practice location was urban for 59 GPs (40%) and 24 pediatricians (86%). Concerning nutrition, four GPs and one pediatrician had a nutrition degree (3% of the 177 physicians). Six GPs and one pediatrician were following a vegetarian diet (5% out of the 177 physicians).

### 3.2. Opinions about the Effects of an OLV or Vegan Diet on Adults and Children

Among adults, most of the physicians considered that the adoption of an OLV diet led to a protective effect on the overall cardiovascular risk, myocardial infarction, high blood pressure, obesity, and global risk of cancer, with a median of 7 on the scores ranging from 0 (harmful effect) to 10 (protective effect) (Table 2). Physicians’ views were less favorable toward the vegan diet in adults, or either vegetarian diet in children.

Regarding the deficiencies, 132 (76%) physicians felt that an OLV diet was at risk of iron deficiency, 84 (49%) mentioned vitamin B12 deficiency and 78 (45%) mentioned protein deficiency (Figure 1). These proportions were higher for a vegan diet: risk of iron deficiency (*n* = 143, 84%), vitamin B12 deficiency (*n* = 124, 69%), and protein deficiency (*n* = 133, 79%) (Figure 2).

### 3.3. Suitability of OLV or Vegan Diets and Reasons for Adopting These Diets

The OLV diet was considered unsuitable for children under six months by 94 physicians (54%), for children over six months by 116 physicians (67%), for teenagers by 93 physicians (53%), for pregnant and breastfeeding women by 87 physicians (51%), and for elderly persons by 88 physicians (51%) (Figure 3). The vegan diet was deemed unsuitable by most of the physicians for all specific subgroups: children under six months (*n* = 148, 87%), children over six months (*n* = 161, 95%), teenagers (*n* = 143, 83%), pregnant and breastfeeding women (*n* = 155, 91%), and elderly persons (*n* = 137, 81%) (Figure 4).

Regarding the reasons why patients chose to follow a vegetarian diet, 86% of the physicians thought that it could be for health reasons (*n* = 150), 80% for animal welfare (*n* = 139), 74% for environmental care (*n* = 130), 52% for fashion (*n* = 89), 32% linked with eating disorder (*n* = 55), and 18% for religious beliefs (*n* = 31).

### 3.4. Vegetarian Diets and Physician–Patient Relationship

Most physicians had seen at least one recognized vegetarian (OLV or vegan diet) patient in the last three months (*n* = 104, 59%). Regarding medical practice, 145 (83%) adapted their routine care for vegetarian children and 88 (51%) for vegetarian adults.

Whereas 24 (14%) physicians declared that they would advise their patients against switching to an OLV diet, 88 (51%) would advise patients against switching to a vegan diet. According to 46% of the physicians (*n* = 81). Most physicians (*n* = 141, 81%) did not feel informed enough about the vegetarian diets (Table 3).

Among GPs, we found determinants associated with advising against a vegan diet. In unadjusted analyses, these factors were GPs aged over 50 years (odds ratio (OR) = 2.1 [95% confidence interval: 1.04–4.2]) and agreeing that a vegan diet is associated with a risk of folate deficiency (OR = 2.2 [1.1–4.4]) (Table 4).

## 4. Discussion

As vegetarian diets are becoming increasingly common in France, the accuracy of information delivered by healthcare providers is crucial. In this survey, while half of the GPs and pediatricians have seen at least one recognized vegetarian patient during the previous three months, most (81%) did not feel sufficiently informed about these diets. Half of the physicians declared that they would dissuade their patient from switching to a vegan diet (51%).

Most of the physicians involved in this survey declared that, in their opinion, OLV and vegan diets were associated with iron deficiency (76% and 84%, respectively) and protein deficiency (45% and 79%, respectively). There is now evidence that balanced vegetarian diets are not at risk of protein deficiency in developed countries [28,29], and although vegetarians usually have lower iron stores, there is no evidence of adverse health effects (moderately lower iron stores may even be associated with a lower risk of chronic diseases) [30]. In our survey, the risk of vitamin B12 deficiency came fourth after iron, calcium, and protein deficiencies, according to the physicians’ beliefs; whereas it represents the greatest risk of deficiency in vegetarian or vegan diets [16]. Vitamin B12 is present only in animal products [31]. Vegan patients, as well as many OLV patients, require vitamin B12 supplementation. Several case reports have been published about vegetarian children with growth disorders due to severe vitamin B12 deficiency [32,33,34], often associated with defective medical follow-up in the context of distrust toward pediatricians by the affected families [35]. An Italian survey showed that parents who chose alternative diets often lack confidence in their pediatrician [36]. In this survey, 22% of the families decided to adopt an alternative diet without pediatric advice. In about half of the cases, families did not consider the pediatrician to be competent enough to give adequate indications regarding their diet. Most of these parents reported antagonism from their pediatrician (77%). These results are consistent with our survey, as 95% of the physicians declared that a vegan diet is not suitable for children. According to another Italian survey conducted in 2020, parents did not inform their primary care pediatricians about the vegan diet in 36% of the cases. In 71% of the cases, primary care pediatricians were perceived as skeptical or against a vegan diet [37].

In univariate analyses, advice against switching to a vegan diet was associated with the age of the GP being older and with the physician believing that vegan diets put patients at risk of folate deficiency. As folate is present in large quantities in plants, OLV and vegan patients are unlikely to have any deficiency. The association with age suggests that younger physicians may be more open or informed to vegetarian and vegan diets. This could be a consequence of recent changes in how vegetarian and vegan diets are considered in the general population [4]. It remains unlikely that recent evolution in the medical education might explain this result, as information on these diets is still scarce throughout medical studies.

Agreement on the risk of folate deficiency in a vegan diet may therefore be interpreted as a marker of limited knowledge in these diets, and such physicians might be more likely to discourage patients from switching to a vegetarian diet. These initial results suggest a lack of awareness and some apprehension of GPs and pediatricians regarding the care of vegetarian patients. Pediatric academic societies have taken different positions worldwide. In France, the Pediatric Hepatology, Gastroenterology and Nutrition Group, advises against a vegan diet in children [38]. Regular dietary monitoring is essential in vegan children, where vitamin D (also noted for other diets) and vitamin B12 supplementation is necessary. Iron, calcium, docosahexaenoic acid, iodine, and zinc can be supplemented on a case-by-case basis.

Concerning adults, they are exposed to vitamin B12 deficiency as well and must be supplemented [16]. Several countries have already published guidelines to help physicians in the follow-up of their vegetarian patients [39,40,41,42]; yet, such recommendations are still lacking in France.

According to the physicians in our survey, patients mainly adopt a vegetarian diet for health reasons (*n* = 150, 86%). In an American survey [43], the authors found that the reason depended on age, and younger people (11 to 20 years old) were more likely to choose a vegetarian diet for moral or environmental reasons. In the above-mentioned Italian survey, the decision to exclude animal foods from the child’s diet was agreed upon by both parents (77.1%); health benefits and ethics were the most common reason (75.5%) [37]. A French survey also found that a vegetarian diet may be a masked form of mental anorexia in a few patients [44]. An American survey studied the inter-relationships between vegetarianism and eating disorders among women. It found that individuals with an eating disorder history were significantly more likely to have ever been vegetarian (52% vs. 12%), to be currently vegetarian (24% vs. 6%), and to be primarily motivated by weight-related reasons (42% vs. 0%) [45]. Knowing this information would allow for earlier detection of these diseases among vegetarians.

Primary care physicians presented a knowledge gap in nutrition; yet, the prevalence of vegetarian diets are increasing. More and more patients choose to follow a vegetarian diet. Physicians perceived that their training was insufficient to deliver good advice to their patient. Specific training in this topic is needed. As stated by the World Health Organization: “Health professionals play a central and critical role in improving access and quality health care for the population. They provide essential services that promote health, prevent diseases and deliver health care services to individuals, families and communities based on the primary health care approach” [46]. Involvement in this area falls not only to physicians alone, but the entire primary care community, i.e., dieticians/nutritionists, psychologists, and nurses.

Our study was limited by a small sample size, and it can be assumed that participating physicians felt more concerned by this subject, which induced recruitment bias. Our population may not be fully representative of French physicians, but we relied on a physician network that is used to conduct epidemiological studies in primary care [27]. As with any survey-based study, our results could have been affected by reporting bias among respondents.

## 5. Conclusions

The results about the medical care of vegetarian patients highlighted an apprehension and a need for information from the GPs and pediatricians. Half of the physicians discouraged their patients from adopting a vegan diet. However, most respondents did not feel sufficiently informed about these diets. They seemed to find themselves helpless when facing this frequent situation, where they are expected to be key for the nutritional education and follow-up of their patient. Specific and detailed recommendations should be made, for example, about the appropriate intake of vitamin B12. It may be of interest to conduct such surveys in countries where recommendations do exist, in order to assess whether physicians are more comfortable with providing advice on these diets.

## Figures and Tables

**Figure 1 nutrients-14-03101-f001:**
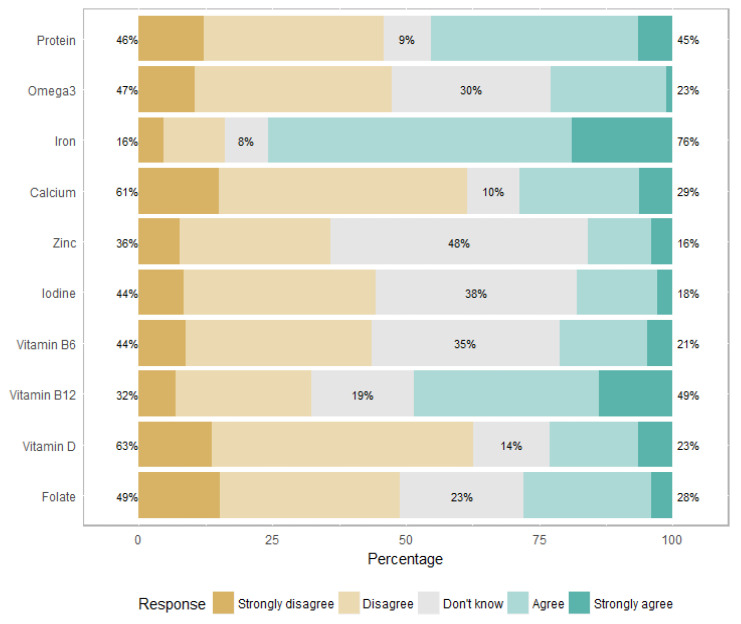
Perceived risk of deficiencies related to an ovo-lacto-vegetarian (OLV) diet.

**Figure 2 nutrients-14-03101-f002:**
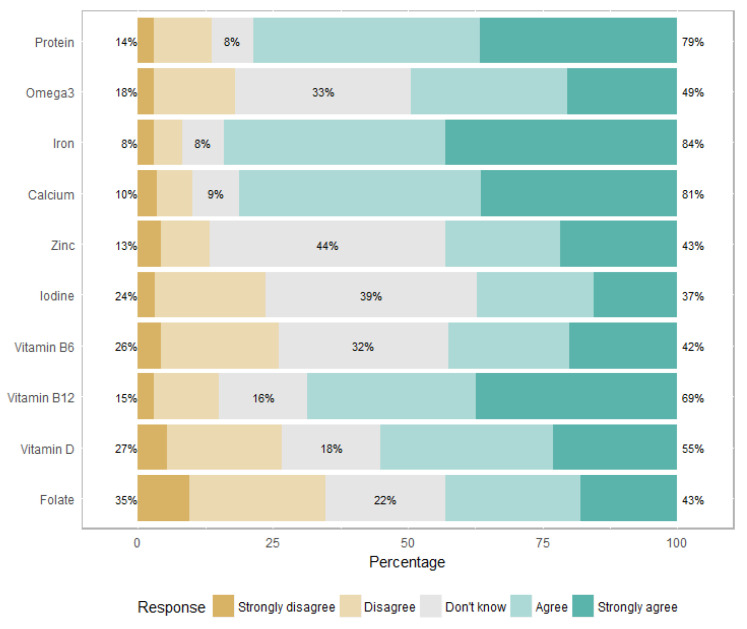
Perceived risk of deficiencies related to a vegan diet.

**Figure 3 nutrients-14-03101-f003:**
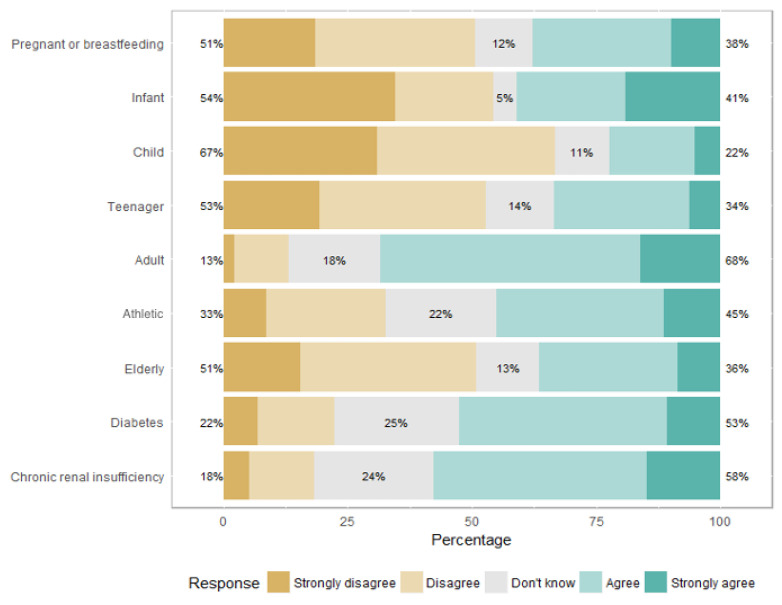
Ovo-lacto-vegetarian diet’s suitability, depending on different populations.

**Figure 4 nutrients-14-03101-f004:**
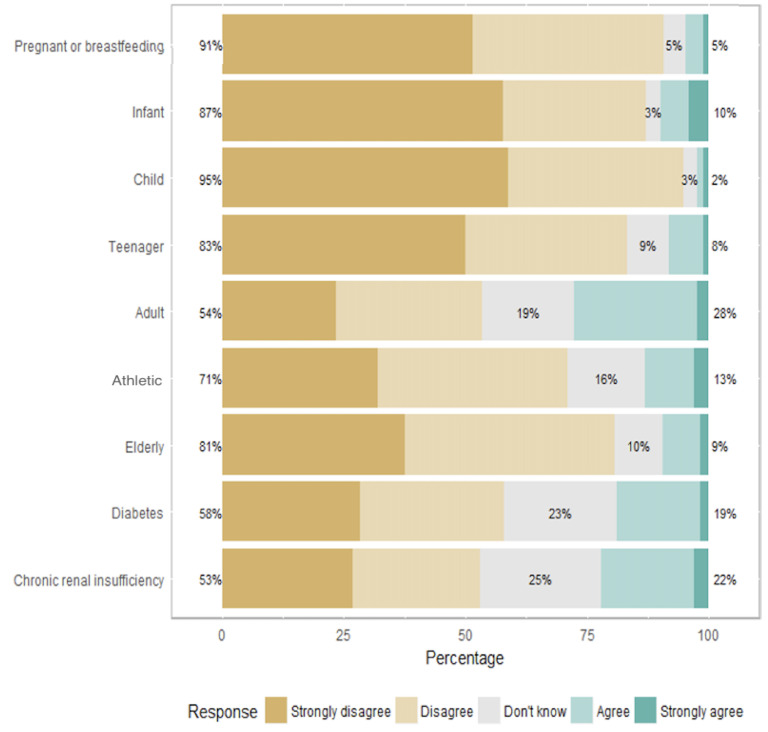
Vegan diet’s suitability, depending on different populations.

**Table 1 nutrients-14-03101-t001:** Characteristics of participating physicians *n* (%).

	Participating GPs N = 149	French GPs * N = 58,450	Participating Pediatricians N = 28	French Pediatricians * N = 2026
Age (median (min–max)), mv: 10	49 (27–71)	51	56 (35–69)	49
Sex, mv: 1				
Female	61 (41)	23,617 (40)	26 (93)	1322 (65)
Male	87 (59)	34,833 (60)	2 (7)	704 (35)
Practice location, mv: 1				
Urban	59 (40)		24 (86)	
Semi-rural	56 (38)		3 (11)	
Rural	33 (22)		1 (3)	
University nutrition degree, mv: 2				
No	143 (97)		26 (96)	
Yes	4 (3)		1 (4)	
Vegetarian diet, mv: 22				
No	124 (95)		24 (96)	
Yes	6 (5)		1 (4)	

* Data from the report on physician demographics 2018 of the Ministry of Solidarity and Health in mainland France (33); mv = missing value.

**Table 2 nutrients-14-03101-t002:** Effects of an ovo-lacto-vegetarian or vegan diet “conducted in the best possible way” on adults and children, according to physicians. (0 = harmful effect; 5 = neutral effect; 10 = protective effect; *n* (%).

	Ovo-Lacto-Vegetarian Diet			Vegan Diet		
Participating Physicians N = 177	Median (Q1–Q3)	I Do Not Know *n*(%)	Missing Values (*n*)	Median (Q1–Q3)	I Do Not Know *n*(%)	Missing Values (*n*)
**Adults**						
Overall cardiovascular disease	7 (5–8)	26 (14)	0	7 (5–8)	39 (23)	9
Myocardial infarction	7 (6–8)	29 (16)	0	7 (5–8)	40 (24)	9
High blood pressure	7 (5–8)	24 (13)	1	6 (5–8)	36 (21)	9
Type 2 diabetes	6 (5–8)	29 (17)	2	6 (5–8)	40 (24)	11
Obesity	7 (6–8)	21 (12)	1	7 (5–8)	33 (20)	12
Cancer (global risk)	7 (5–8)	34 (19)	2	6 (5–7)	46 (27)	10
Osteoporosis	5 (3–5)	25 (15)	2	2 (1–4)	24 (14)	9
Anemia	3 (2–5)	14 (8)	0	2 (1–3)	15 (9)	9
Neurodegenerative disorder	5 (5–6)	67 (39)	7	5 (3–5)	71 (43)	13
Depression	5 (4–5)	63 (37)	7	4 (2–5)	60 (36)	11
**Children (2–11 years old)**						
Obesity in childhood	6 (5–8)	36 (20)	1	7 (5–8)	48 (29)	9
Obesity in adulthood	7 (5–8)	43 (25)	2	6 (5–8)	50 (30)	9
Anemia	3 (2–5)	14 (8)	0	2 (1–3)	15 (9)	7
Growth disorder	4 (2–5)	24 (14)	1	2 (1–3)	16 (9)	7
Psychomotor development disorder	5 (3–5)	31 (18)	3	3 (1–4)	36 (21)	7
Asthma	5 (5–5)	62 (36)	7	5 (4–5)	73 (45)	15

**Table 3 nutrients-14-03101-t003:** Vegetarian diets and physician–patient relationships *n* (%).

	Total Participating Physicians N = 177	Participating GPs N = 149	Participating Pediatricians N = 28	Missing Values
Vegetarian patient seen in the last three months				2
Yes	104 (59)	90 (61)	14 (50)	
No	64 (37)	51 (35)	13 (46)	
I do not know	7 (4)	6 (4)	1 (4)	
Dissuasion when switching to an OLV diet				2
Absolutely	2 (1)	1 (1)	1 (4)	
Prefer yes	22 (13)	13 (9)	9 (32)	
Prefer not	70 (40)	56 (38)	14 (50)	
Absolutely not	81 (46)	77 (52)	4 (14)	
I do not know	0 (0)	0 (0)	0 (0)	
Dissuasion when switching to a vegan diet				2
Absolutely	38 (22)	23 (16)	15 (54)	
Prefer yes	50 (29)	39 (27)	11 (39)	
Prefer not	58 (33)	57 (39)	1 (3,5)	
Absolutely not	27 (15)	27 (18)	0 (0)	
I do not know	2 (1)	1 (1)	1 (3,5)	
Sufficient information about vegetarian diets				3
Absolutely	6 (3)	6 (4)	0 (0)	
Prefer yes	26 (15)	19 (13)	7 (25)	
Prefer not	83 (48)	72 (49)	11 (39)	
Absolutely not	58 (33)	48 (33)	10 (36)	
I do not know	1 (1)	1 (1)	0 (0)	
Physician-patient relationship influenced by patient’s diet choice				1
Absolutely	7 (4)	5 (3)	2 (7)	
Prefer yes	74 (42)	60 (41)	14 (50)	
Prefer not	59 (34)	52 (35)	7 (25)	
Absolutely not	25 (14)	23 (16)	2 (7)	
I do not know	11 (6)	8 (5)	3 (11)	
Adult-specific care				3
Yes	88 (51)	71 (48)	17 (65)	
No	45 (26)	43 (29)	2 (8)	
I do not know	41 (23)	34 (23)	7 (27)	
Adult-specific biological follow-up				5
Yes	102 (59)	87 (59)	15 (60)	
No	43 (25)	40 (27)	3 (12)	
I do not know	27 (16)	20 (14)	7 (28)	
Child-specific care				2
Yes	145 (83)	120 (82)		
No	11 (6)	11 (7)		
I do not know	19 (11)	16 (11)		
Child-specific biological follow-up				3
Yes	128 (74)	103 (70)	25 (89)	
No	16 (9)	14 (10)	2 (7)	
I do not know	30 (17)	29 (20)	1 (4)	

GP: general practitioner; OLV: ovo-lacto-vegetarian.

**Table 4 nutrients-14-03101-t004:** Determinants associated with advising against a vegan diet in the GP population N = 149.

Variables	Advising against a Vegan Diet		OR in Univariate Analysis [95% CI]
	Yes *n* (%)	No *n* (%)	
**Socio-Demographic Criteria**			
Age			
50 years old	24 (33)	48 (67)	ref
>50 years old	33 (51)	32 (49)	2.1 [1.04–4.2]
Sex			
Male	36 (42)	50 (58)	ref
Female	26 (43)	34 (57)	1.1 [0.5–2.1]
Exercise environment			
Urban	22 (39)	35 (61)	ref
Semi-rural	26 (46)	30 (54)	1.4 [0.7–2.9]
Rural	14 (42)	19 (58)	1.2 [0.5–2.8]
Nutrition degree			
Yes	1 (25)	3 (75)	ref
No	60 (43)	81 (57)	2.2 [0.3–45.5]
Vegetarian Diet followed by physicians			
Yes	2 (33)	4 (66)	ref
No	55 (45)	67 (55)	1.6 [0.3–12.2]
Vegetarian patient seen in the last 3 months			
Yes	40 (45)	49 (55)	ref
No	18 (35)	33 (65)	0.7 [0.3–1.3]
I do not know	3 (50)	3 (50)	1.2 [0.2–6.9]
Insufficient information on vegetarian diets			
Yes	9 (47)	16 (53)	ref
No	53 (52)	67 (48)	1.4 [0.6–3.6]
**Perceived risk of deficiency**			
Folate if vegan diet			
Disagree/I do not know	30 (35)	56 (65)	ref
Agree	29 (53)	25 (46)	2.2 [1.1–4.4]
Folate if OLV diet			
Disagree/I do not know	40 (38)	66 (62)	ref
Agree	20 (54)	17 (46)	1.9 [0.9–4.2]
Vitamin B12 if vegan diet			
Disagree/I do not know	14(31)	31 (69)	ref
Agree	45 (48)	49 (52)	2.0 [0.97–4.4]
Vitamin B12 if OLV diet			
Disagree/I do not know	27 (36)	48 (64)	ref
Agree	34 (49)	35 (51)	1.7 [0.9–3.4]

Ref: reference category; OLV: ovo-lacto-vegetarian; OR: odds ratio.

## Data Availability

The data are available on request from the Sentinelles network www.sentiweb.fr (accessed on 27 July 2022).

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
