# Peer review of "Vegetarian and Vegan Diets: Beliefs and Attitudes of General Practitioners and Pediatricians in France"

_nutrients, 2022, doi:10.3390/nu14153101_

Round 1

Reviewer 1 Report

The topic and purpose of the manuscript are very important. Overall, I consider the article a highly valuable piece of work. The authors have elaborated on a particularly current topic.

In the Introduction chapter, the authors presented the important pieces of literature required for the topic of the study. The number of references collected in the manuscript is minimal. The existing sections should be supplemented with additional literature. In particular, the authors should collect and analyze literature that describes the preference of general practitioners and paediatricians for plant-based nutrition. Furthermore, it would be worthwhile to conduct detailed searches to see how a plant-based diet and related counseling can arise during the doctor-patient relationship.

The authors used descriptive statistics and other multivariate statistics in their research to a usual standard in the article. The presentation of results in the tables is presented in a clear and relevant way.

Chapter 4 is one of the greatest strengths of the manuscript. The authors compare their own results with those of other similar research to a high standard.

The authors minimally explain the limitations related to the research, the detailing of this and the description of the further directions of the research should be a very important part of the manuscript.

Author Response

The topic and purpose of the manuscript are very important. Overall, I consider the article a highly valuable piece of work. The authors have elaborated on a particularly current topic.

In the Introduction chapter, the authors presented the important pieces of literature required for the topic of the study. The number of references collected in the manuscript is minimal. The existing sections should be supplemented with additional literature. In particular, the authors should collect and analyze literature that describes the preference of general practitioners and paediatricians for plant-based nutrition. Furthermore, it would be worthwhile to conduct detailed searches to see how a plant-based diet and related counseling can arise during the doctor-patient relationship.

We thank the reviewer for this comment. Introduction has been reworked. We added 5 new references. We added in particular this paragraph:

“A recent systematic review showed that a vegan diet may result in deficiencies in micronutrients (vitamin B12, zinc, calcium and selenium) which should not be disregarded [20]. Another systematic review comparing qualitatively the vegetarian and nonvegetarian diets conclude that OVL or vegans had higher overall diet quality (4.5-16.4 points higher on the Healthy Eating Index 2010 [HEI-2010]) compared with nonvegetarians in 9 of 12 studies [21]. Authors explained that, for vegetarians, higher scores were driven by closer adherence to recommendations for total fruit, whole grains, seafood and plant protein, and sodium. For nonvegetarians, adherence to recommendations for refined grains and total protein foods were better.

Diet is central to improve health outcomes. Yet, nutrition counseling worldwide is very limited during medical visits [22, 23]. Barriers have been identified as limited time and knowledge gaps [22, 24]. Concerning knowledge, a systematic review showed that nutrition is insufficiently incorporated into medical education, regardless of country, setting, or year of medical education [25]. Deficits in nutrition education affect students' knowledge, skills, and confidence to implement nutrition care into patient care. A study evaluated how US medical students and family medicine residents rated their nutrition training [26]. Most trainees felt they were inadequately trained during medical school although they expressed high interest in whole-foods plant-based nutrition to address the burden of chronic diseases [26].”

The authors used descriptive statistics and other multivariate statistics in their research to a usual standard in the article. The presentation of results in the tables is presented in a clear and relevant way. Chapter 4 is one of the greatest strengths of the manuscript. The authors compare their own results with those of other similar research to a high standard.

Thank you for your support.

The authors minimally explain the limitations related to the research, the detailing of this and the description of the further directions of the research should be a very important part of the manuscript.

In order to respond to this comment, we added following paragraph:

“Primary care physicians presented a knowledge gap in nutrition, Yet, vegetarian diets are increasing. More and more patients choose to follow a vegetarian diet. Physician perceived that their training is insufficient to deliver good advices to their patient. Specific training in this topic is needed. As stated by World Health Organization: “Health professionals play a central and critical role in improving access and quality health care for the population. They provide essential services that promote health, prevent diseases and deliver health care services to individuals, families and communities based on the primary health care approach.[46]” Not only the physician but the entire primary care community has to be involved in this area, i.e. dietician/nutritionist, psychologist, nurse.”

“Our study was limited by small sample size and it can be assumed that participating physicians felt more concerned by this subject, which induced recruitment bias. Our population may not be fully representative of the French physicians but we relied on a physician network that is used to conduct epidemiological studies in primary care {27]. As with any survey-based study, our results could have been affected by reporting bias among respondents.”

Reviewer 2 Report

Villette et al, present an interesting study on the views and beliefs of physicians regarding plant-based diets in France. The manuscript is well written and flows well. A few points that could improve the manuscript are:

1) Please clarify what was included as a nutrition degree

2) Lines 147-148: "the physician-patient relationship might be affected by the diet choice of their patients." Please remove this phrase from the results. You could report statements like this in the discussion section as a possible explanation and provide the appropriate reference if available.

3) In the limitation section of the manuscript please include the fact that the survey was conducted via Questionnaires and this could affect the accuracy of the results.  

4) Please include in your introduction or conclusion systematic reviews that examined the adequacy of vegetarian/vegan diets so as provide the readers with scientific data and not only describe the situation in France. 

Author Response

Villette et al, present an interesting study on the views and beliefs of physicians regarding plant-based diets in France. The manuscript is well written and flows well. A few points that could improve the manuscript are:

1) Please clarify what was included as a nutrition degree

We clarified this point and added, in the method section, the following paragraph:

“In the survey, nutrition degree included university diploma or interuniversity di-ploma in the field of nutrition.”

2) Lines 147-148: "the physician-patient relationship might be affected by the diet choice of their patients." Please remove this phrase from the results. You could report statements like this in the discussion section as a possible explanation and provide the appropriate reference if available.

We thank the reviewer for this comment and removed this sentence from the result. Discussion according to the other remarks has also been reworked.

3) In the limitation section of the manuscript please include the fact that the survey was conducted via Questionnaires and this could affect the accuracy of the results.  

This limitation has been added.

4) Please include in your introduction or conclusion systematic reviews that examined the adequacy of vegetarian/vegan diets so as provide the readers with scientific data and not only describe the situation in France. 

Introduction and discussion had been reworked. For example, we added:

“A recent systematic review showed that a vegan diet may result in deficiencies in micronutrients (vitamin B12, zinc, calcium and selenium) which should not be disregarded [20]. Another systematic review comparing qualitatively the vegetarian and nonvegetarian diets conclude that OVL or vegans had higher overall diet quality (4.5-16.4 points higher on the Healthy Eating Index 2010 [HEI-2010]) compared with nonvegetarians in 9 of 12 studies [21]. Authors explained that, for vegetarians, higher scores were driven by closer adherence to recommendations for total fruit, whole grains, seafood and plant protein, and sodium. For nonvegetarians, adherence to recommendations for refined grains and total protein foods were better.

Diet is central to improve health outcomes. Yet, nutrition counseling worldwide is very limited during medical visits [22, 23]. Barriers have been identified as limited time and knowledge gaps [22, 24]. Concerning knowledge, a systematic review showed that nutrition is insufficiently incorporated into medical education, regardless of country, setting, or year of medical education [25]. Deficits in nutrition education affect students' knowledge, skills, and confidence to implement nutrition care into patient care. A study evaluated how US medical students and family medicine residents rated their nutrition training [26]. Most trainees felt they were inadequately trained during medical school although they expressed high interest in whole-foods plant-based nutrition to address the burden of chronic diseases [26].”

Reviewer 3 Report

The manuscript entitled ‘Vegetarian and vegan diets: beliefs and attitudes of general  practitioners and paediatricians in France’ presents interesting issue, however some corrections are needed.

The English language should be improved (e.g. it should be ‘pediatricians’’ instead of ‘paediatricians’ line 3) 

Lines 42-43 – ‘… and red meat, which is associated to an increased risk of cancer (especially 43 colorectal and stomach) [11,12]. – please add ‘especially of processed meat’ [please see Cancer: Carcinogenicity of the consumption of red meat and processed meat. https://www.who.int/news-room/questions-and-answers/item/cancer-carcinogenicity-of-the-consumption-of-red-meat-and-processed-meat.]

Introduction section is quite short. The good background should present the history of problem, the current knowledge and scientific "gap", and then authors should present how their study could fill this gap to justify the study. I do not see any justification – please consider it. 

Authors should in their discussion include 3 areas: (1) compare gathered data with the results by other authors, (2) formulate implications of the results of their study and studies by other authors, (3) formulate the future areas which should be studied.

Authors should present here and discuss the limitations of their study. The limitation section should be at the end of the discussion scion. 

The subject of the study is very important. The knowledge of physician related to a nutritional counseling (diet recommendation) is often not enough. So there is an urgent need to not only improved it, but also to create the team (physician, dietician/ nutritionist, and psychologist) to maintain and improved patient's health. 

Author Response

The manuscript entitled ‘Vegetarian and vegan diets: beliefs and attitudes of general  practitioners and paediatricians in France’ presents interesting issue, however some corrections are needed.

– The English language should be improved (e.g. it should be ‘pediatricians’’ instead of ‘paediatricians’ line 3) 

We thank the reviewer for this comment and we corrected the article.

– Lines 42-43 – ‘… and red meat, which is associated to an increased risk of cancer (especially 43 colorectal and stomach) [11,12]. – please add ‘especially of processed meat’ [please see Cancer: Carcinogenicity of the consumption of red meat and processed meat. https://www.who.int/news-room/questions-and-answers/item/cancer-carcinogenicity-of-the-consumption-of-red-meat-and-processed-meat.]

We changed as recommended.

– Introduction section is quite short. The good background should present the history of problem, the current knowledge and scientific "gap", and then authors should present how their study could fill this gap to justify the study. I do not see any justification – please consider it. 

We thank the reviewer for this comment. Introduction has been reworked. We added in particular this paragraph:

“A recent systematic review showed that a vegan diet may result in deficiencies in micronutrients (vitamin B12, zinc, calcium and selenium) which should not be disregarded [20]. Another systematic review comparing qualitatively the vegetarian and nonvegetarian diets conclude that OVL or vegans had higher overall diet quality (4.5-16.4 points higher on the Healthy Eating Index 2010 [HEI-2010]) compared with nonvegetarians in 9 of 12 studies [21]. Authors explained that, for vegetarians, higher scores were driven by closer adherence to recommendations for total fruit, whole grains, seafood and plant protein, and sodium. For nonvegetarians, adherence to recommendations for refined grains and total protein foods were better.

Diet is central to improve health outcomes. Yet, nutrition counseling worldwide is very limited during medical visits [22, 23]. Barriers have been identified as limited time and knowledge gaps [22, 24]. Concerning knowledge, a systematic review showed that nutrition is insufficiently incorporated into medical education, regardless of country, setting, or year of medical education [25]. Deficits in nutrition education affect students' knowledge, skills, and confidence to implement nutrition care into patient care. A study evaluated how US medical students and family medicine residents rated their nutrition training [26]. Most trainees felt they were inadequately trained during medical school although they expressed high interest in whole-foods plant-based nutrition to address the burden of chronic diseases [26].”

– Authors should in their discussion include 3 areas: (1) compare gathered data with the results by other authors, (2) formulate implications of the results of their study and studies by other authors, (3) formulate the future areas which should be studied.

We thank the reviewer for this comment. Discussion has been reworked. We added in particular this paragraph:

“Primary care physicians presented a knowledge gap in nutrition, Yet, vegetarian diets are increasing. More and more patients choose to follow a vegetarian diet. Physician perceived that their training is insufficient to deliver good advices to their patient. Specific training in this topic is needed. As stated by World Health Organization: “Health pro-fessionals play a central and critical role in improving access and quality health care for the population. They provide essential services that promote health, prevent diseases and deliver health care services to individuals, families and communities based on the primary health care approach.[46]” Not only the physician but the entire primary care community has to be involved in this area, i.e. dietician/nutritionist, psychologist, nurse.”

– Authors should present here and discuss the limitations of their study. The limitation section should be at the end of the discussion scion. 

The limitations are at the end of the discussion:

Our study was limited by small sample size and it can be assumed that participating physicians felt more concerned by this subject, which induced recruitment bias. Our population may not be fully representative of the French physicians but we relied on a physician network that is used to conduct epidemiological studies in primary care {27]. As with any survey-based study, our results could have been affected by reporting bias among respondents.”

– The subject of the study is very important. The knowledge of physician related to a nutritional counseling (diet recommendation) is often not enough. So there is an urgent need to not only improved it, but also to create the team (physician, dietician/ nutritionist, and psychologist) to maintain and improved patient's health. 

Thank you for your support.

Round 2

Reviewer 3 Report

I appreciate the great efforts that the authors have made in response to my questions and concerns. I have no further comments